# The Translational Role of MUC8 in Salivary Glands: A Potential Biomarker for Salivary Stone Disease?

**DOI:** 10.3390/diagnostics11122330

**Published:** 2021-12-10

**Authors:** Martin Schicht, Adrian Reichle, Mirco Schapher, Fabian Garreis, Benedikt Kleinsasser, Malik Aydin, Afsun Sahin, Heinrich Iro, Friedrich Paulsen

**Affiliations:** 1Institute of Functional and Clinical Anatomy, Friedrich Alexander University Erlangen-Nürnberg, 91054 Erlangen, Germany; adrian.reichle@t-online.de (A.R.); fabian.garreis@fau.de (F.G.); benedikt.kleinsasser@fau.de (B.K.); 2Department of Otorhinolaryngology, Head and Neck Surgery, FAU Medical School, Friedrich Alexander University Erlangen-Nürnberg, 91054 Erlangen, Germany; Mirco.Schapher@klinikum-nuernberg.de (M.S.); Heinrich.Iro@uk-erlangen.de (H.I.); 3Laboratory of Experimental Pediatric Pneumology and Allergology, Center for Biomedical Education and Research, School of Life Sciences (ZBAF), Department of Human Medicine, Faculty of Health, Witten/Herdecke University, 58448 Witten, Germany; Malik.Aydin@uni-wh.de; 4Center for Child and Adolescent Medicine, Center for Clinical and Translational Research (CCTR), Helios University Hospital Wuppertal, Witten/Herdecke University, 42283 Wuppertal, Germany; 5Department of Ophthalmology, Koc University Medical School, Sarıyer, Istanbul 34450, Turkey; afsunsahin@gmail.com; 6Department of Operative Surgery and Topographic Anatomy, Sechenov University, 119991 Moscow, Russia

**Keywords:** salivary gland, salivary stone, saliva, mucin, MUC8, TNFα, diseases 1

## Abstract

Mucin (MUC) 8 has been shown to play an important role in respiratory disease and inflammatory responses. In the present study, we investigated the question of whether MUC8 is also produced and secreted by salivary glands and whether it may also play a role in the oral cavity in the context of inflammatory processes or in the context of salivary stone formation. Tissue samples from parotid and submandibular glands of body donors (*n* = 6, age range 63–88 years), as well as surgically removed salivary stones from patients (*n* = 38, age range 48–72 years) with parotid and submandibular stone disease were immunohistochemically analyzed targeting MUC8 and TNFα. The presence of MUC8 in salivary stones was additionally analyzed by dot blot analyses. Moreover, saliva samples from patients (*n* = 10, age range 51–72 years), who had a salivary stone of the submandibular gland on one side were compared with saliva samples from the other “healthy” side, which did not have a salivary stone, by ELISA. Positive MUC8 was detectable in the inter- and intralobular excretory ducts of both glands (parotid and submandibular). The glandular acini showed no reactivity. TNFα revealed comparable reactivity to MUC8 in the glandular excretory ducts and also did not react in glandular acini. Salivary stones demonstrated a characteristic distribution pattern of MUC8 that differed between parotid and submandibular salivary stones. The mean MUC8 concentration was 71.06 ng/mL in female and 33.21 ng/mL in male subjects (*p* = 0.156). Saliva from the side with salivary calculi contained significantly (15-fold) higher MUC8 concentration levels than saliva from the healthy side (*p* = 0.0005). MUC8 concentration in salivary stones varied from 4.59 ng/mL to 202.83 ng/mL. In females, the MUC8 concentration in salivary stones was significantly (2.3-fold) higher, with an average of 82.84 ng/mL compared to 25.27 ng/mL in male patients (*p* = 0.034). MUC8 is secreted in the excretory duct system of salivary glands and released into saliva. Importantly, MUC8 salivary concentrations vary greatly between individuals. In addition, the MUC8 concentration is gender-dependent (♀ > ♂). In the context of salivary stone diseases, MUC8 is highly secreted in saliva. The findings support a role for MUC8 in the context of inflammatory events and salivary stone formation. The findings allow conclusions on a gender-dependent component of MUC8.

## 1. Introduction

One of the most common inflammatory diseases of the salivary glands is sialolithiasis (salivary stone formation) [1,2,3,4]. In this context, calcified concretions usually form in the salivary gland excretory duct system of the salivary glands. The main symptom is pain, due to partial or complete obstruction in the excretory duct system and associated gland swelling [1,2,3,4]. Most commonly, sialoliths occur in the major paired salivary glands, with the submandibular gland being the most affected, accounting for approximately 80% of cases [1,2,3,4]. This is attributed in part to the anatomy of Wharton’s duct and in part to salivary flow vs. gravity, both of which counteract a continuous salivary flow. About 20% of stones occur in the parotid gland, and beyond that very rarely in the sublingual gland and the minor salivary glands [5,6]. The process of sialolith formation is still not fully understood, and different theories have been discussed [7,8,9,10,11,12]. Recently, neutrophil extracellular traps were reported to participate in sialolithogenesis, providing a promising explanation for salivary stone formation by uniting different theories in a common final path, which is based on an inflammatory process [13].

Biochemical analyses have shown that sialoliths consist of organic and inorganic material with great variation in the relative contribution [1,14]. Submandibular stones contain approximately 9–12% and parotid stones up to 20% organic material by their dry weight [1,4,15]. The organic parts consist of collagen, neutral and acid glycoproteins, other proteins, lipids, and carbohydrates such as glucose and mannose [1,2,3,4]. Using immunoblotting techniques, a large, unidentified glycoprotein could be identified in solubilized submandibular sialoliths, as well as some lower molecular weight proteins [16]. Studying dacryoliths (mucopeptide concretions from the lacrimal sac), Paulsen et al. (2006) observed that major organic components are mucins (MUC) and trefoil factor peptides (TFF) [17]. Here, MUC8 was the only mucin (with one exception) that could always be detected.

Mucins are the main component of the mucus, are high molecular weight epithelial glycoproteins, and can be divided into two structurally and functionally distinct classes: the secreted type and the membrane-associated type. The number of studies of mucins in salivary glands and saliva are rather small relative to studies of other glands and secretions. Mannweiler et al. (2003) described MUC1 and MUC2 in the ductal epithelium of healthy parotid glands, but Kutta et al. could only detect MUC1, not MUC2 [18,19]. MUC7 was detected in submandibular and sublingual gland saliva, but not in parotid saliva [19,20]. Jagla et al. (1999) noted that mucins were rarely detected in the parotid gland [21]. Wickström et al. (2000) identified MUC5B and MUC7 in saliva of the oral cavity [22]. These authors assumed that the prevalence of mucins in parotid saliva was minimal. Mucinous cells of the submandibular and sublingual glands and some minor salivary glands have been shown to be glandular sources of MUC5B [23,24,25,26,27]. MUC7 is produced by mucinous and serous components of the submandibular, sublingual, and some minor glands [24,26,27,28]. As mentioned earlier, MUC8 has been found to be regularly depleted in dacryoliths [17]. Surprisingly, there has been no evidence of salivary stones, salivary glands, and saliva associated with MUC8.

MUC8 was first isolated in submucosal glands of the human trachea in 1994. The gene is located on chromosome 12q24.33 [29,30]. The total cDNA of MUC8, unlike that of other mucins, has not been completely deciphered to date; hence, the functions and properties of MUC8 are not fully elucidated at present. Furthermore, studies on genetically modified laboratory animals have not yet been possible [31]. However, MUC8 is known to be upregulated in the context of inflammatory processes in the respiratory tract and in the pathogenesis of salivary stone diseases; inflammatory events also occur regularly [32]. It is known that the regulatory effects of cytokines such as TNFα and IL-1β synergistically increase mucin secretion [33]. 

Based on the described properties of MUC8 and the fact that MUC8 has been shown to be a regular constituent of dacryoliths [17], the aim of this study was to determine the possible occurrence of MUC8 in the two major salivary glands, Glandula (Gld.) parotidea and Gld. submandibularis, in saliva and in salivary stones from both glands, in order to draw possible conclusions about the composition and pathophysiology of salivary stones.

## 2. Results

### 2.1. Localization of MUC8 in Submandibular and Parotid Glands

In principle, all performed immune reactions worked with both antibodies. Both MUC8 and TNFα produced very similar results. MUC8 was visible in both the parotid and the submandibular gland in all major intralobular excretory ducts, which had a distinct lumen (Figure 1a,b). All cells of the single-layered epithelium of the excretory ducts reacted positively with the antibody; individual cells reacted particularly intensely. MUC8 showed clear reactivity in the submandibular gland at the luminal cell pole of the epithelial cells, whereas individual epithelial cells reacted only weakly or not at all (Figure 1a insert). In the parotid gland, a higher intensity was visible around individual cell nuclei (Figure 1b). MUC8 reactivity was not detectable in either the mucosal or serous acini of the submandibular gland, which occupied the majority of the tissue section. The same was true for the exclusively serous terminals of the parotid gland. In both tissues, single positively labeled cells outside the excretory ducts were observable (Figure 1a,b: arrows with asterisk). In these cells, the reaction of the entire cytoplasm was usually homogeneously positive. The cells could not be clearly assigned to histological structural associations. In Figure 1b, these cells were located in close proximity to serous acini, but an affiliation to the acini could not be determined by cell overlays.

### 2.2. Localization of TNFα in Submandibular and Parotid Glands

After successful immunoreaction with the antibody against MUC8, an immunoreaction was performed with an antibody against TNFα using the same protocol on the same tissue samples. This should show whether there are parallels between the distribution of MUC8 and the inflammatory mediator TNFα, as known from the literature [33]. Antibody detection of TNFα yielded a picture corresponding to that of MUC8 localization. The epithelia of the intralobular excretory ducts showed specific immune reactions. The difference from the result with MUC8 in the submandibular gland was that here TNFα was also partially detectable with a luminal increase, but also the whole cytoplasm of the cells contained the protein (Figure 1c). In the parotid gland, there was also a homogeneous distribution over entire cells; a slight increase in reactivity around the cell nucleus could be observed in isolated cases (Figure 1d).

### 2.3. Localization of MUC8 and TNFα in Salivary Stones from Submandibular Gland

Examination of salivary stones from the submandibular gland showed a layered structural composition. However, there were individual differences in the structure of the stones. The stones examined with the antibody targeting MUC8 provided radially arranged layers, some of which were arranged around what were thought to be several centers of their own (Figure 2a). With TNFα, the majority of the layers were arranged around a common center, reminiscent of the annual rings of trees (Figure 2c). Both TNFα and MUC8 occurred in individual layers distributed throughout the stone. Some of the antibody responses were of high and some of low intensities. The distribution of the two proteins was very similar. No logical pattern could be discerned. In Figure 2c, a slight increase in reactivity from an imaginary center to the periphery could be observed in the lower third of the image. Close inspection of the layers revealed a punctate, granule-like arrangement in the individual layers for both proteins (see insert Figure 2a,c). Furthermore, the stones were riddled with artificial cracks. In addition, Figure 2c also showed some cloudy, amorphous areas interrupting the layer pattern. A distinct band of cell nuclei could also be observed here. These two aspects were not seen in Figure 2a, where the layered pattern extended through the entire section, interrupted only by cracks. Larger amorphous areas and cell nuclei were not visible.

### 2.4. Localization of MUC8 and TNFα in Salivary Stones from Parotid Gland

The stones from the parotid gland had an unstructured morphology (Figure 2b,d). A stratification as described for stones from the submandibular gland was missing. Instead, there were areas that reacted more strongly and areas that reacted less intensely with antibodies against MUC8 and TNFα. There were positive MUC8 reactions almost in the whole stone, which appeared as reddish shading (Figure 2b). Amorphous areas, which had a crystal-like structure, showed weak or no positive signals. A band of strong MUC8 intensity was visible, and at higher magnification, nuclei contained within it could be seen (see insert Figure 2b). In contrast, however, the majority of the nucleated areas were devoid of MUC8 signal. Salivary stone treated with TNFα showed predominantly crystalline or glassy areas with no positive responses (Figure 2d). A line of strong intensity ran along the edge of the stone and also within the stone, containing individual cell nuclei. At the lower right part of the image, larger clusters of cell nuclei were observable, again showing no positive responses. TNFα could be detected in parts that did not contain a strong crystalline structure on one side and clear portions of cellular material on the other side (see insert Figure 2d).

### 2.5. MUC8 Is Detectable in All Salivary Stones

For the detection of MUC8 in salivary stones, protein samples obtained from the salivary stones were used. Twenty-four different salivary stones from the excretory duct system of the mandibular salivary gland of 24 different patients were examined; untreated saliva diluted in TBST from 3 subjects of the working group served as positive control. MUC8 was detectable in all used samples. The evaluation showed that there were significant differences between the individual samples, which could be seen in the intensity (blackening) of the spot formed on the PVDF membrane. The clearer the blackening, the higher was the concentration of contained protein. Individual samples showed a high content of MUC8 (Figure 3: samples 2, 5, 10, 20, 24—intense blackening), whereas in other samples less MUC8 was detectable (Figure 3: e.g., samples 1, 16—weak blackening). The positive controls also showed different intensities.

### 2.6. MUC8 ELISA of Submandibular Salivary Stones

Fourteen salivary stones from the submandibular glands were analyzed. As already shown by means of dot blot, MUC8 was detectable in all salivary stones. The MUC8 concentrations determined ranged from 4.69 ng/mL to 202.827 ng/mL (see Table 1), resulting in an average value of 86.341 ng/mL (±18.35). The mean age of the patients was 49 years (±3.370), of whom ten were males and four females. Statistical analysis of the distribution by gender revealed significantly higher MUC8 concentrations in female compared to male patients (*p* ≤ 0.038) (Figure 4a). In female patients, the mean MUC8 concentration was 144.7 ng/mL (±38.30), and in male patients, 62.99 ng/mL (±16.52). The MUC8 levels were 2.3-fold higher in female patients. With respect to age, no increased occurrence of MUC8 in salivary stones could be detected in those over or under 50 years of age (*p* ≤ 0.8178) (Figure 4b). Six of the patients were under 50 years of age, whereas eight were over 50 years of age. The mean values of the two groups were insignificantly different: 79.79 ng/mL (±26.99) (<50 years) and 91.26 ng/mL (±26.37) (>50 years). The distribution was based on the average age of the patients and was chosen in favor of a balanced number of values.

### 2.7. MUC8 Is More Abundant in Females

Examination of the saliva samples yielded MUC8 concentrations ranging from 0.419 ng/mL to 143.952 ng/mL, with an average value of 46.86 ng/mL (±16.56) (see Table 2, sample 3943r not detectable). The saliva samples were obtained from paired glands of patients, each of whom had unilateral sialolithiasis. There were six male and four female patients; the average age was 63 years. Statistical analysis of the distribution by gender revealed significantly higher MUC8 concentrations in female patients compared to male patients (*p* ≤ 0.034; Figure 5a). No significant differences resulted from the evaluation of the values with respect to age (see Figure 5b). The mean concentration of MUC8 was 82.84 ng/mL (±27.38) in female patients and 25.27 ng/mL (±10.11) in male patients. Patients who were over 60 years old showed a mean MUC8 level of 39.94 ng/mL (±15.33), whereas patients who were under 60 years old showed a mean of 59.05 ng/mL (±21.36). Again, the cutoff of 60 years had been set based on the mean age of the patients in order to have a more equal distribution of values for statistical analysis. When comparing healthy (no salivary stone present) and diseased (salivary stone present) submandibular glands, significantly increased MUC8 concentrations were seen in glands with salivary stone (*p* ≤ 0.0005; Figure 6). MUC8 was increased approximately 15-fold in glands with salivary stone; the mean value for glands without salivary stone was 2.528 ng/mL (±7.183) and 74.58 ng/mL (±15.99) for glands with salivary stone.

## 3. Discussion

To date, no exact etiology for the development of salivary stones is known, but there are some hypotheses discussed in the literature [1]. It has been observed in several patient cases that salivary stones developed around an organic foreign body [7,34]. This led to the so-called “retrograde theory of salivary stone formation” [8], which states that food debris, bacteria, or other organic substances can enter the excretory ducts of salivary glands and become nuclei for calcification and deposition. In addition, in dacryoliths of the lacrimal sac, cores with a well-defined structure (nidus) were recently described using a new 3D reconstruction technique (Cinematic rendering; Siemens Healthineers AG, Erlangen, Germany) from computed tomographic (CT) scans, accounting for approximately 10% of the total stone [35]. Another hypothesis holds microsialoliths responsible as the cause of salivary stones. These microstones have been detected intracellularly in acini, interstitially, and in the lumina of salivary gland ducts. It has been shown in a post-mortem cohort of patients that 80% of the submandibular glands have microstones on microscopic examination [2]. In the acinar cells, they could be identified as autophagosomes formed as a product of autophagocytosis of calcium-rich secretory granules and cell organelles. Accumulation of calcium ions and cell organelles in the phagosomes eventually leads to precipitation of calcium and thus calcification [2]. After release into the glandular ducts, the microliths represent nucleation centers for further deposition and calcification. As the microliths increase in size as a result, micro obstructions may develop in portions of the glandular ducts, thereby locally reducing salivary flow. In a recent study using a number of different techniques, Schaper et al. (2020) hypothesized that neutrophil extracellular traps (NETs) are involved in sialolithogenesis and that the majority of sialoliths harbor bacterial DNA. At the very least, this study suggests that NETs initiate the formation and growth of sialoliths in humans. Deposition of extracellular DNA from neutrophil granulocytes around small crystals leads to their dense aggregation, and subsequent mineralization produces alternating dense mineral layers composed predominantly of calcium salt deposits and DNA [13]. Thus, further precipitation of calcium and other salivary components is triggered, and further stone growth is indicated. In the case of an existing salivary stone, it has been observed that further interlobular microstones are responsible for the maintenance and continuation of inflammatory changes in the affected gland, leading to fibrosis of the affected salivary gland [36].

Our present results clearly show that MUC8 is produced by excretory duct epithelial cells of the parotid and submandibular glands and is also released into saliva (Figure 1, Figure 3 and Figure 5). Since saliva samples from non-stone-bearing as well as stone-bearing salivary glands contain MUC8, MUC8 can be understood as a permanent salivary constituent. In this context, it will be interesting in the future to shed light on the further functional significance of MUC8 in the oral cavity, e.g., with regard to pellicle formation and gingival health as well as bacterial colonization. The finding that MUC8 is regularly present in salivary stones is consistent with the finding of Proctor et al. (2005), who used immunoblotting techniques to regularly identify a large undefined glycoprotein in submandibular stones [16]. The main finding, that significantly higher MUC8 concentrations are found in the saliva of stone-bearing glands, allows the hypothesis that MUC8 is increasingly expressed in connection with the events occurring during stone formation (Figure 6). Particularly because sialolithiasis is associated with a recurrent inflammatory component and MUC8 is increasingly secreted in the context of inflammatory processes, the findings that MUC8 is deposited externally (submandibular gland) or diffusely (parotid gland) in the form of “annual rings” on the salivary stone nuclei and the significantly higher MUC8 concentration in salivary stones are of great importance.

In the context of the inflammatory component, Seong et al. (2002) demonstrated that the expression of MUC8 is increased in human nasal polyps compared with other mucins and that in the presence of inflammatory mediators such as TNFα and IL-1β, the mRNA of MUC8 is increased in nasal epithelial cells [32], as previously shown for normal human nasal epithelial cells by Yoon et al. (1999) [33]. Our finding that TNFα had a MUC8-like distribution in the studied salivary gland stones strongly supports this inflammatory hypothesis of increased MUC8 content in salivary stones. Similarly, IL-1β has been shown to cause increased secretion of MUC8 in ciliary epithelial cells [37]. A similar relationship has also been described in epithelial cells of the middle ear. The presence of IL-1β or the presence of otitis media is associated with higher MUC8 concentrations and a higher density of ciliary-bearing cells in the epithelium [38]. In allergic reactions and associated overproduction of mucus, IL-13 is implicated. In vitro studies of cultured nasal epithelial cells have shown that the secretion of MUC8 can be increased by IL-13 [39]. MUC8 thus plays a role in the development and course of various types of chronic sinusitis, from which polyps (polyposis nasi) can develop secondarily. Lee et al. showed that in the mucosa of patients with chronic rhinosinusitis, the expression of MUC8 is increased compared to its levels in healthy subjects [40]. Elevated levels of MUC8 have also been detected in lower respiratory tract and lung diseases. MUC8 is present in mucosal gland cells of the trachea and bronchi in patients with cystic fibrosis [41]. At the gene level, a link between MUC8 and chronic obstructive pulmonary disease (COPD) has been demonstrated [42]. Further studies to understand the exact relationships between the physiology and function of MUC8, also in combination with other mucins, to ultimately create a potential therapeutic benefit, e.g., in COPD [43], are lacking. A first approach in this direction is the suggestion that MUC8 has anti-inflammatory properties. In studies in which MUC8 was knocked down by using siRNA, an increase in inflammatory mediators such as IL-1α and IL-6 was detected, and at the same time a reduction was noted in anti-inflammatory cytokines such as TGF-β and IL-1α receptor antagonist. This process is coupled to the ATP-dependent P2Y2 receptor, which stimulates the release of inflammatory mediators in the context of airway inflammation. Activation of the receptor simultaneously causes increased release of MUC8 [31,44].

The increase in expression of MUC8 is coupled to signal transduction chains involving MAP kinases [45,46,47,48,49,50,51,52,53]. Here, proinflammatory mediators such as prostaglandin E2 or visfatin can cause elevated MUC8 levels, highlighting the role of MUC8 in the context of inflammatory events [48,49,50]. Furthermore, oxygen free radicals are known to lead to respiratory diseases. Oxygen radicals are released by, among others, NADPH oxidase (abbreviated NOX). It has been shown that PDGF is increased in the mucosa in the presence of sinusitis and induces the generation of oxygen free radicals via NOX, thereby increasing the expression of MUC8 [54]. Similar effects are elicited by cadmium, a toxic metal found in exhaust fumes and cigarette smoke, among others, which can induce airway inflammation [46]. All of these findings suggest that MUC8 is not only upregulated by proinflammatory cytokines but acts as an anti-inflammatory antagonist. This hypothesis would very well explain the observed “annual rings” in the submandibular stones. However, further studies are needed in this regard.

Interestingly, the MUC8 concentration in saliva is about twice as high in women as in men (Figure 5). Evidence for hormonal regulation of salivary gland proteins is available from a few animal studies [55,56]. According to these studies, both sex hormones (estrogens, androgens) as well as thyroid hormones have an influence on certain salivary gland proteins. It will be interesting to investigate the hormonal influence of these and other hormones with respect to MUC8 concentration in the future. No statement can be made that women have more severe cases of salivary stones. The level of suffering can vary extraordinarily between the sexes on an individual basis. For example, patients may have very small stones and a high level of clinical symptoms or, conversely, very large stones without an enormous level of suffering. In addition, we are able to write that the level of MUC8/TNFa expression does not correlate with the weight/size of the stone. We always measured the same amounts of stone tissue via ELISA, which were consistent with each other in the analysis, and in fact, there were clear sub-individual variations here. It was only obvious that women on average have higher concentrations than men.

In connection with the formation of submandibular calculi, our study showed an increased MUC8 concentration by a factor of 15 in the saliva of the stone-affected side, which is another clear indication of the inflammatory process (Figure 2). MUC8 should therefore be further analyzed as an interesting marker for the occurrence of salivary stone disease (and also for the formation of dacryoliths).

Limitations of the present study include the small sample size and the older age of the body donors. Although the samples were carefully selected from cadavers that had no history of nasal or oral cavity disease, there is a possibility that involutional changes may have influenced the results. In addition, we investigated whether MUC 8 could be a diagnostic marker for salivary gland stones. As already discussed, it is well-known that MUC8 has an important role in mucus hypersecretion in chronic sinusitis [32]. Fluctuating MUC8 expression in saliva could deregulate mucus expression and alter the rheological properties of saliva. Such altered viscosity could in turn be a reason for salivary stone formation [57]. However, whether the increased MUC8 level in salivary stones should be regarded as the cause of salivary stone formation or whether it occurs in response to the presence of a salivary stone cannot be answered with these results yet, which is a study limitation. Possibly, and we assume this on the basis of the stone morphology in the form of annual rings and our other results, both factors influence each other. We only examined MUC8 in the present work. To be absolutely sure that we have identified a central MUC in the context of salivary stone formation, other mucins identified in saliva and salivary glands, such as at least MUC5B and MUC7, would also have to be controlled for completeness, which is another study limitation. Although we can verify the function of the antibodies used against MUC8 and TNFα with control tissues as stated, it is of course a major drawback of our study that there was no control tissue for the examination of the stones to directly compare the expression of the two proteins with male and female tissue. Another study limitation is that we neither measured nor weighed the stones during surgical removal. For this reason, no information can be given on this. We also did not perform ELISA studies for TNFα, so we cannot add these ELISA data, either.

In conclusion, our results demonstrate a role for MUC8 in relation to inflammatory events and salivary stone formation. Our results suggest a sex-dependent component of MUC8.

## 4. Materials and Methods

### 4.1. Tissues, Sialoliths, and Saliva

The study was conducted in compliance with institutional review board regulations, informed consent regulations, and the provisions of the Declaration of Helsinki and its later amendments. A written informed consent was obtained from each patient concerning diagnostic procedures, treatment to remove salivary stones, and the subsequent use of these specimens for research purposes, including data analysis, prior to inclusion in the study. The study was approved by the University’s ethical review committee (186_19 Bc).

#### 4.1.1. Salivary Gland Tissue 

Salivary gland samples were used for histological and immunohistochemical examination. The tissue samples (from 6 submandibular and 2 parotid glands) were from 8 body donors (submandibular: 3 males, 3 females; age range: 63–88 years; and parotid: 1 male, 1 female, aged between 73 and 88 years) donated to the Institute of Anatomy, FAU Erlangen-Nürnberg, Erlangen, Germany. Samples were taken from the body donors within 24 h post mortem, fixed in 4% paraformaldehyde (PFA), and subsequently embedded in paraffin. Donors were free of recent trauma, oral or tooth infections, and diseases involving or affecting salivary function. Nasal mucosa from the area of the ethmoidal bone with described MUC8 protein production (Jung et al., 2000) and thymus tissue (for TNF) were used as control tissues and included in each study batch.

#### 4.1.2. Sialoliths

Seven salivary stones from the excretory ducts of the parotid gland, and 31 salivary stones from submandibular glands from a total of 38 patients (see below) were provided from the Department of Otorhinolaryngology, Head and Neck Surgery of the University Hospital of FAU Erlangen-Nürnberg, Germany. Of these, 2 sialoliths from the parotid and 4 from the submandibular glands were selected for immunohistochemical examination and fixed in 4% PFA solution immediately after surgical removal. After fixation, these 6 salivary stones were demineralized in Ethylenediaminetetraacetic acid (EDTA) solution for 14 days to allow histological sections to be prepared after embedding in paraffin. The remaining 32 stones (5 parotid, 27 submandibular) were stored at −80 °C for further purposes. Finally, not all sialoliths could be included in the investigations. One of the parotid stones was too small to extract sufficient protein from it; the other parotid stone dissolved completely during decalcification. The same was true for some sialoliths from the submandibular gland. Their inconsistent nature made processing impossible in individual cases. Thus, in total only 24 sialoliths from the submandibular gland could be included in the investigations. 

All patients in our study cohort with symptomatic sialolithiasis (*n* = 38; 22 men, 16 women; mean age 48.7 and 23.3 years) underwent clinical examination, ultrasound imaging, and subsequent sialendoscopy to confirm the diagnosis. Diagnostic and therapeutic procedures were performed at a tertiary referral center specializing in salivary gland diseases (FAU Medical School, Department of Otolaryngology, Head and Neck Surgery, University of Erlangen-Nürnberg, Erlangen, Germany), and stones were obtained from submandibular (*n* = 31) and parotid glands (*n* = 7).

Submandibular stones were removed by transoral surgery, which allowed obtaining a complete unfractionated sialolith. Parotid stones were removed either by an open surgical approach, in which the stone was removed as a whole, or by sialendoscopically assisted basket extraction (if the calculus was small enough to evacuate through the canal). For larger parotid concrements that could not be removed as a whole due to of their size or that could not be obtained by an open surgical approach, stone fragments were collected after sialendoscopically guided intraductal pneumatic lithotripsy followed by basket extraction.

#### 4.1.3. Saliva

Two saliva samples were obtained from each of 10 patients at the ENT clinic of the University Hospital of FAU Erlangen-Nürnberg with stone disease on one side. One sample per patient was extracted from the right or from the left submandibular gland. A salivary stone was found in one excretory duct of the two glands. Saliva was collected using a capillary tube that was placed into the opening of the sublingual caruncle on the corresponding side. Collected saliva samples were immediately frozen and stored at −80 °C.

### 4.2. Immunohistochemistry

For immunohistochemistry, reactivity was followed with an antibody against the mucin peptide core epitope of MUC8 and an antibody against TNF in tissue sections (7 μm) from the parotid and submandibular glands. The antibody targeting MUC8 was studied on sections subjected to 10 min of microwave heating pretreatment as described previously [17]. Primary antibodies against MUC8 (Proteintech, Manchester, UK, 55489-1-AP, 1:250, polyclonal) and TNF (Santa Cruz Biotechnology, Inc., Dallas, TX, USA, 52B83, 1:50, monoclonal) were used. All primary antibodies were applied overnight at room temperature (RT). Secondary antibodies goat anti-rabbit IG (1:200, Dako, Glostrup, Denmark) or goat anti-mouse (1:200; Dako, Glostrup, Denmark) were incubated for at least 4 h at RT. Visualization was performed with peroxidase-labeled streptavidin-biotin for at least 5 min. After counterstaining with hemalum, sections were mounted in Aquatex (Boehringer, Germany). Two negative control sections were used in each case; one was incubated with secondary antibody only, the other with primary antibody only. Nasal epithelium and thymus, respectively, were used as positive controls. All slides were examined with a microscope (Keyence, BZ9000, Osaka, Japan).

### 4.3. Protein Isolation and Determination of Protein Concentration from Salivary Stones

For the molecular biological examination of the salivary stones, the proteins were first extracted and quality controlled. For this purpose, the stored salivary stones frozen at −80 °C were placed in lysis tubes, each filled with 300 μL Triton buffer and 3 μL protease inhibitor plus phosphatase inhibitor. If the stones were smaller than 3–4 mm, only half of the indicated volumes were used. The stone samples were crushed in a Speedmill (Analytik Jena GmbH, Jena, Germany), then incubated on ice for 30 min and finally placed in a refrigerated centrifuge for 30 min. The centrifuge was operated at 13,000 rpm at 4 °C. The supernatants were transferred to Eppendorf cups and stored at −80 °C until further work. To determine the content of proteins, the Bradford method was used according to manufacturer’s instructions.

### 4.4. Visualization of MUC8 by Immunoblot Analysis

A PVDF membrane prewetted with TBST was placed on a 96-well dot blot apparatus (Biometra GmbH, Göttingen, Germany). From each salivary stone sample, 25 μg was mixed with 50 μL TBST. As a control, 3 slots were loaded with 50 μL saliva in 50 μL TBST. The slots were rinsed again with 50 μL TBST after application. The membrane was then swirled in 5% milk powder solution in TBST buffer for 1 h at RT and incubated in a humidity chamber with the primary antibody against MUC8 (polyclonal, 55489-1-AP, rabbit anti-human, Proteintech, St. Leo-Rot, Germany; diluted 1:100 in 5% milk powder solution) overnight at 4 °C. Subsequently, the secondary antibody (goat anti rabbit IgG HRP, 1:2000, Santa Cruz, CA, USA) was incubated for 2 h at RT. After a final rinse, an enhanced chemiluminescence (ECL) mix was incubated in a darkened Eppendorf cup at a 1:1 ratio to visualize MUC8 by reacting it with the HRP (horse radish peroxidase) residue of the secondary antibody, emitting light. The PVDF membrane was placed in a dark chamber, and the ECL mix was incubated on it for 5 min. Finally, the membrane was placed in a Universal Hood II instrument (Bio-Rad Laboratories, Feldkirchen, Germany) and evaluated by a built-in camera that can detect the emitted light. A picture of the membrane was taken every 5 min for 30 min. The dot blot served us as a very simple detection method to indicate whether MUC8 was detectable in all stones.

### 4.5. Enzyme-Linked Immunosorbent Assay (ELISA)

ELISA analysis was performed using a human MUC8 protein kit (Human Mucin-8, MUC8 ELISA Kit, MBS9305835, quantitative sandwich, MyBiosource, San Diego, CA, USA) and the appropriate protocols from MyBiosource.com. Analysis was performed using a microplate spectrophotometer (ELISA reader ClarioStar, BMG Labtech GmbH, Munich, Germany) at wavelengths of 405 nm and 450 nm to measure the absorbance of saliva samples. By comparison with the standard series and the determined values for antigen concentration (protein concentration), absorbance of each sample was calculated in ng/mg.

### 4.6. Statistical Analysis

The results of the ELISA test were statistically evaluated using the program Graph Prism (version 5.01, GraphPad Software Inc., San Diego, CA, USA) and graphically displayed as standard error of the mean (SEM). For statistics, the Kolmogorov–Smirnov test and Kruskal–Wallis test analysis procedures were used. A *p*-value smaller than ≤0.05 was set as statistically significant and was marked with an asterisk (*) in the graph.

## 5. Conclusions

MUC8 is a physiological secretion product of the excretory duct system of the major salivary glands and varies individually in its concentration. However, women have significantly higher MUC8 salivary concentrations than men. Increased formation and secretion of MUC8 into saliva occurs in the context of salivary stone disease. Our results support a role for MUC8 in the context of inflammatory events in salivary stone formation, making MUC8 an interesting candidate biomarker for salivary stone disease. Of particular interest in this context is the sex-dependent component of MUC8, which should also be further elucidated in the context of other stone diseases.

## Figures and Tables

**Figure 1 diagnostics-11-02330-f001:**
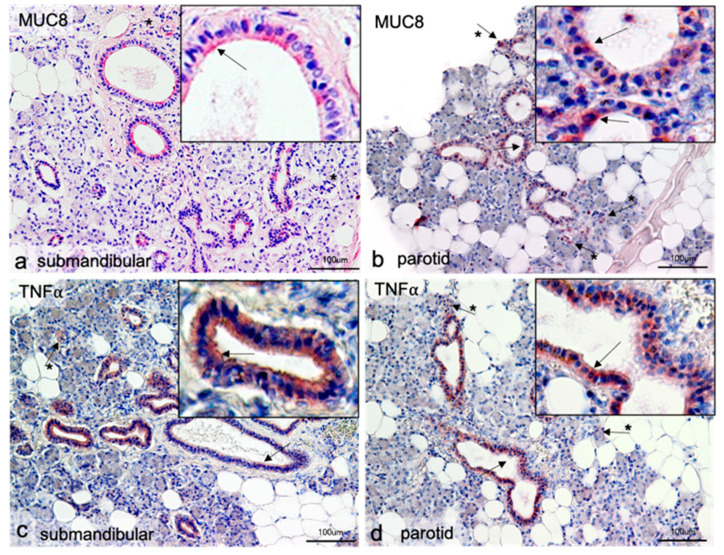
Immunohistochemical localization of MUC8 (**a**,**b**) and TNFα (**c**,**d**) in submandibular and parotid glands. In all gland sections, positive antibody responses are visible in the epithelial cells of the intralobular excretory ducts (arrows). The antibody against MUC8 reacts in the submandibular gland at the luminal cell pole of the epithelial cells (insert **a**); in the parotid, gland single more intense reactions around cell nuclei are visible (insert in **b**). The antibody reactions against TNFα give similar pictures. Here, more intense signals around cell nuclei are also visible in the Gld. parotidea (insert **d**). In the Gld. submandibularis, no clear polarization can be observed for TNFα; mostly the whole cytoplasm reacts homogeneously, but there are also single cells with more intense reactions (insert **c**). Furthermore, single positive cells are seen in the glandular parenchyma in all tissues (arrows with asterisk). Neither serous nor mucosal acini show positive antibody responses.

**Figure 2 diagnostics-11-02330-f002:**
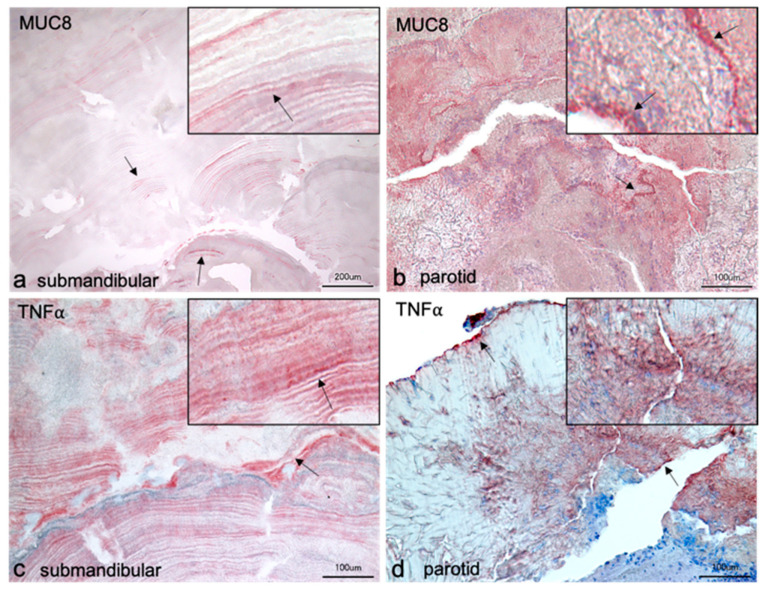
Immunohistochemical localization of MUC8 (**a**,**b**) and TNFα (**c**,**d**) in salivary stones from the excretory ducts of the submandibular and parotid glands. The sections of the submandibular stones show layered arrangements. The structure in **a** is comparable to the switching lamellae of bone; different areas show radially arranged layers around supposedly their own centers. In contrast, the structure in **c** is reminiscent of the annual rings of trees, as here the layers appear to be arranged around a single center. The antibody against MUC8 reacts positively in individual layers, but alternately shows no signals in other layers. In layers with stronger reactions, punctate granule-like signals are observable (arrows and insert in **a**). These granule-like signals in the layers are also visible with the antibody against TNFα (arrows and insert in **c**). In the lower third of the image, a slight increase in reactivity of the red-stained layers is visible in **c** from bottom to top until a band of blue-stained cell nuclei is seen. No positive signals for TNFα occur in the region of the cell nuclei. In the remaining part of the stone, very distinct layers are seen interspersed with cloud-like amorphous parts. Positive MUC8 reactions occur almost throughout the section of the parotid stone (**b**). A band of stronger reactivity can be seen, containing cell nuclei (arrows and insert in **b**). The positive MUC8 signals appear as cloudy shading. Areas with crystalline structure do not show positive responses. The parotid stone in **d** shows predominantly crystalline structures, which also do not give positive signals. Intense bands at the stone edge and within the stone with intercalated cell nuclei are visible (arrows in **d**).

**Figure 3 diagnostics-11-02330-f003:**
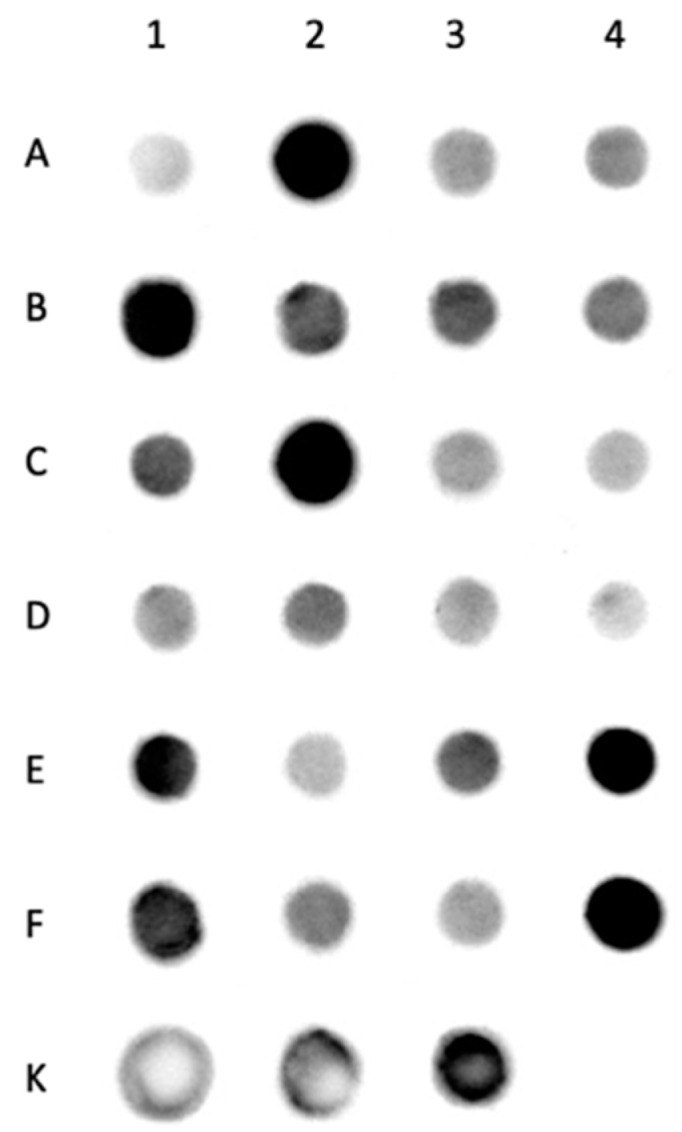
Dot blot analysis of MUC8 in salivary stones from the excretory ducts of the submandibular glands. Each dot shows the positive detection of MUC8 in a salivary stone. The darker the dot, the higher the MUC8 concentration. Next to it in tabular form the sample designations, 1–24 denote the protein samples of the salivary stones, K1–K3 describe the saliva samples as positive controls. The intensity of the black staining indicates the different amounts of MUC8 contained. Individual samples (2, 5, 10, 20, 24) show a strong signal, whereas others react only very weakly (e.g., 1, 16). The positive controls are also stained black with varying intensity.

**Figure 4 diagnostics-11-02330-f004:**
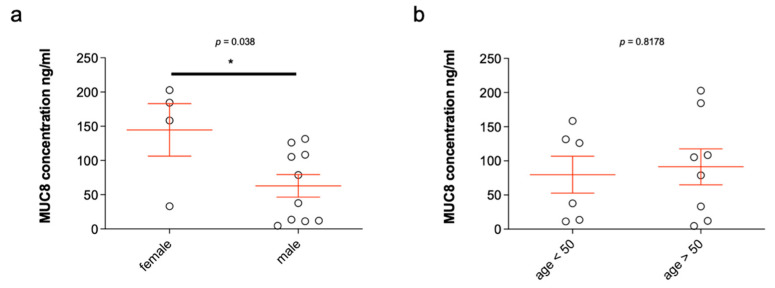
MUC8 concentrations in salivary stones from the excretory ducts of the submandibular glands in relation to sex (**a**) and age (**b**). The concentrations of MUC8 are shown in ng/mL for each single stone analyzed (*n* = 14, Table 1). The MUC8 concentrations shown in the bar chart (**a**) illustrate the large scatter of concentration values between stones (Table 1). Statistically, the MUC8 concentrations determined are significantly higher in female patients (**a**; *p* ≤ 0.038). When considering MUC8 values in over and under 50-year-old patients, no significant difference can be detected (**b**; *p* ≤ 0.8178) (* *p* ≤ 0.05).

**Figure 5 diagnostics-11-02330-f005:**
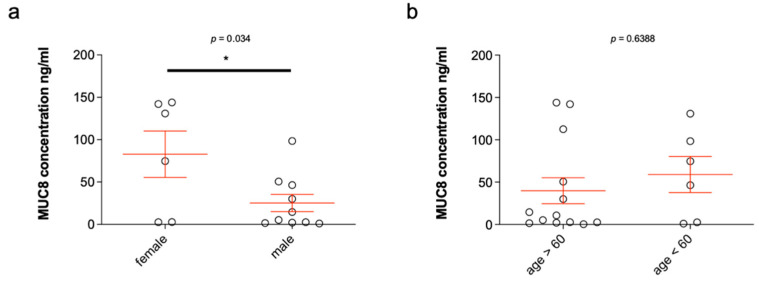
MUC8 concentrations in 20 saliva samples from the excretory ducts of the submandibular glands of 10 patients who had a salivary stone in the excretory duct system of the submandibular gland on one side (x) and not on the other (Table 2) and statistical analysis in relation to sex (**a**) and age (**b**). The concentrations of MUC8 are shown in ng/ml in each case. The MUC8 concentrations shown in the bar chart (Table 2) illustrate the large scatter of concentration values between the individual saliva samples. Statistical analysis of the determined MUC8 concentrations shows statistically significant values in the distribution by gender (**a**, *p* ≤ 0.034) and no significant values for distribution by age (**b**, Table 2) (* *p* ≤ 0.05).

**Figure 6 diagnostics-11-02330-f006:**
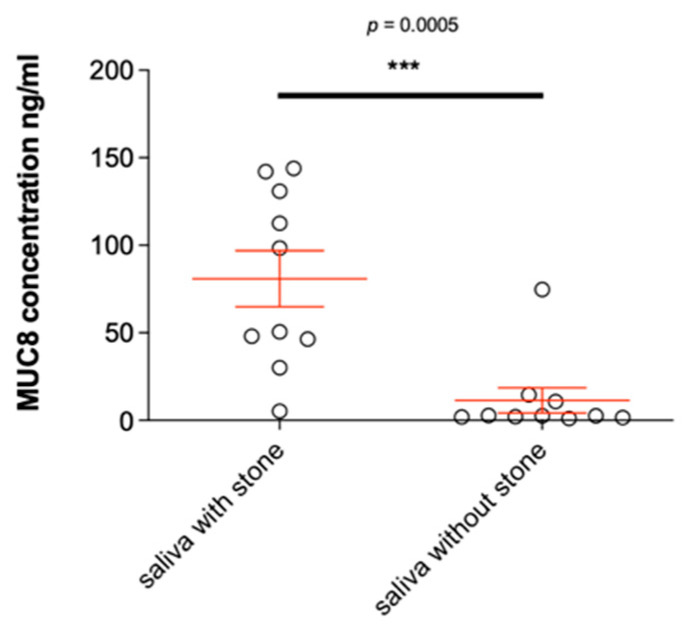
Representation of the different MUC8 concentrations of saliva samples from submandibular glands with and without salivary stone in the excretory duct system. The concentration of MUC8 is indicated in ng/mL. There is a statistically highly significant difference in MUC8 concentration of saliva with salivary stone (*n* = 10) as opposed to without salivary stone (“healthy”, *n* = 10, *p* ≤ 0.0005). MUC8 is significantly increased in salivary glands with stone in the excretory duct system. (*** *p* < 0.001).

**Table 1 diagnostics-11-02330-t001:** Overview of MUC8 concentrations in salivary stones (m = males; f = females).

Salivary Stone No.	MUC8 Concentration in ng/mL	Patient Age at Collection	Sex
1	12.09	72	m
2	184.568	62	f
3	33.141	52	f
4	4.69	51	m
5	13.588	48	m
6	126.208	42	m
7	105.362	55	m
8	78.833	58	m
9	158.369	25	f
10	11.258	34	m
11	131.549	44	m
12	37.755	31	m
13	202.827	53	f
14	108.531	54	m

**Table 2 diagnostics-11-02330-t002:** Overview of MUC8 concentrations in saliva samples (m = male; f = female).

Sample	MUC8 Concentration in ng/mL	Salivary Stone	Patient Age at Collection	Sex
r1	74.813		59	f
l1	130.928	x		
r2	50.666	x	62	m
l2	14.565			
r3	2.531		51	m
l3	46.494	x		
r4	1.561		64	m
l4	112.571	x		
r5	30.051	x	66	m
l5	2.05			
r6	5.284	x	73	m
l6	10.769			
r7	2.749		61	f
l7	143.952	x		
r8	2.525		60	f
l8	142.044	x		
r9	98.49	x	59	m
l9	1.026			
r10	1.994		72	f
l10	48.113	x		

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
