# Peer review of "The Translational Role of MUC8 in Salivary Glands: A Potential Biomarker for Salivary Stone Disease?"

_diagnostics, 2021, doi:10.3390/diagnostics11122330_

Round 1

Reviewer 1 Report

The authors well studied the role that MUC8 might have with regard to salivary stones. They studied it in many ways. Most of the aspect are well assessed, but only one aspect was not studied nor discussed. Namely, was MUC8 present and a possible cause that a salivary stone was developed or was the elevated expression of MUC8 a result of the fact that a salivary stone developed or was present. This should be discussed as well.

Author Response

Point-to-point response

Dear Editor in Chief Ms. Zoey Guo and Guest Editor Prof. Omar Cauli

we sincerely thank you for the opportunity to revise our manuscript. We have made every effort to address all concerns expressed by the two reviewers. All changes are highlighted in the manuscript and a detailed rebuttal letter is attached.

We believe that the manuscript has been significantly improved based on the reviewers' comments. If there are any outstanding questions, please do not hesitate to let us know. We thank you very much for your time. We look forward to hearing from you on how our manuscript is progressing.

Yours sincerely,

Martin Schicht and Friedrich Paulsen

Reviewer 1:

The authors well studied the role that MUC8 might have with regard to salivary stones. They studied it in many ways. Most of the aspect are well assessed, but only one aspect was not studied nor discussed. Namely, was MUC8 present and a possible cause that a salivary stone was developed or was the elevated expression of MUC8 a result of the fact that a salivary stone developed or was present. This should be discussed as well.

We thank the reviewer for this comment and have taken up the point in the end of the discussion:

“We investigated whether MUC 8 could be a diagnostic marker for salivary gland stones. It is well known that MUC8 has an important role in mucus hypersecretion in chronic sinusitis (Seong et al. 2009). Fluctuating MUC8 expression in saliva could deregulate mucus expression and alter the rheological properties of saliva. Such altered viscosity could in turn be a reason for salivary stone formation (Afanas et al. 2003). On the other hand, however, we cannot ultimately say with certainty from our studies whether the increased MUC8 level in salivary stones should be regarded as the cause of salivary stone formation or whether it occurs in response to the presence of a salivary stone. This must be stated as a clear study limitation. Possibly, and we assume this on the basis of the stone morphology in the form of annual rings and our other results, both factors influence each other.”

Reviewer 2 Report

The present study by Schicht et al discusses the role of MUC8 in sialolithiasis and proposes it as a biomarker. The authors observed elevated levels of MUC8 in both the salivary stone and saliva of patients with salivary stones. Mucins, especially MUC8 has been reported previously to be elevated in airway inflammation and in dacryoliths. Overall, the manuscript is well written, and the results supports the hypothesis proposed. However, there are some major concerns that needs to be addressed before acceptance.

Comments:

  1. Line 76: ‘Here, MUC8 was the only mucin (with one exception) that could always be detected

It is unclear why did the authors selected MUC8, as the author mentioned this has been previously observed in dacryoliths. Whereas other mucins such as MUC5B and MUC7 has been detected in salivary glands/saliva, did the authors check the status of some of these mucins to conclude that MUC8 can be regarded as the biomarker?

  1. Immunolocalization of MUC8 and TNFa:

One of the major drawbacks of this study is lack of control tissues to better understand the expression of MUC8 and TNFa in control male / female tissues. The authors will need to address this in the limitation of this study section.

  1. Line 392: ‘4.1. Tissues, dacryoliths and saliva’

Is this a typing error? This is confusing, did the authors collected dacryoliths (naso-lacrymal stones) or sialoliths (salivary stones), did the authors performed expression studies using lachrymal stone or salivary stone?

  1. Though the sample size is very small, the authors observed higher MUC8 expression in females, this is a very interesting and novel observation. However, epidemiological evidence show either a higher prevalence in males or equal prevalence in male and females (https://www.nature.com/articles/sj.bdj.2014.1054 ). Do the authors observe more severity of the salivary stones in females? is higher MUC8 / TNFa expression related to the weight/size of the stone? It will be informative for the reader to include the stone weight/size in table 1. What is the status of salivary TNFa concentration in samples in table 2.

  1. Figure 3 describes the dot blot of MUC8 expression in 24 salivary sones. I did not find any information for these 24 samples in the manuscript. The authors need to mention the sample details as a table. In addition, a densitometric analysis with respect to a control gene is required to achieve a statistical conclusion.

Author Response

Point-to-point response

Dear Editor in Chief Ms. Zoey Guo and Guest Editor Prof. Omar Cauli

we sincerely thank you for the opportunity to revise our manuscript. We have made every effort to address all concerns expressed by the two reviewers. All changes are highlighted in the manuscript and a detailed rebuttal letter is attached.

We believe that the manuscript has been significantly improved based on the reviewers' comments. If there are any outstanding questions, please do not hesitate to let us know. We thank you very much for your time. We look forward to hearing from you on how our manuscript is progressing.

Yours sincerely,

Martin Schicht and Friedrich Paulsen

Reviewer 2:

the present study by Schicht et al. discusses the role of MUC8 in sialolithiasis and proposes it as a biomarker. The authors observed elevated levels of MUC8 in both the salivary stone and saliva of patients with salivary stones. Mucins, especially MUC8 has been reported previously to be elevated in airway inflammation and in dacryoliths. Overall, the manuscript is well written, and the results supports the hypothesis proposed. However, there are some major concerns that needs to be addressed before acceptance.

Comments:

  1. Line 76: ‘Here, MUC8 was the only mucin (with one exception) that could always be detected’

It is unclear why did the authors selected MUC8, as the author mentioned this has been previously observed in dacryoliths. Whereas other mucins such as MUC5B and MUC7 has been detected in salivary glands/saliva, did the authors check the status of some of these mucins to conclude that MUC8 can be regarded as the biomarker?

We thank the reviewer for pointing this out and agree with her/his comment. We did not analyze the salivary stones, but we examined the production and the concentration of MUC5B and MUC7. The reasons for this are mentioned in the introduction: Mannweiler et al. (2003) described MUC1 and MUC2 in the ductal epithelium of healthy parotid glands, but Kutta et al. could only detect MUC1, not MUC2 [18, 19]. MUC7 was detected in submandibular and sublingual gland saliva, but not in parotid saliva [19, 20]. Jagla et al. (1999) noted that mucins were rarely detected in the parotid gland [21]. Wickström et al. (2000) identified MUC5B and MUC7 in saliva of the oral cavity [22]. These authors assumed that the prevalence of mucins in parotid saliva was minimal. Mucinous cells of the submandibular and sublingual glands and some minor salivary glands have been shown to be glandular sources of MUC5B [23-27]. MUC7 is produced by mucinous and serous components of the submandibular, sublingual, and some minor glands [24, 26-28].

Based on these findings from these previous studies and the fact that we examined both parotid and submandibular stones and found MUC8, we assumed that MUC7 and MUC5B might not be of significant importance with respect to stone formation. This finding is also consistent with our findings in lacrimal duct stones. Finally, Proctor et al. using immunoblotting techniques detected only a single large, unidentified glycoprotein in solubilized submandibular sialoliths, as well as in addition some lower molecular weight proteins [16]. This finding also led us to believe that MUC8 might be this glycoprotein and contributed to our assumption that no other mucins play a significant role. Nevertheless, we agree with the reviewer that other mucins would still need to be checked. We have therefore included this point under limitations:

“We only examined MUC8 in the present work. To be absolutely sure that we have identified a central mucin in the context of salivary stone formation, other mucins identified in saliva and salivary glands, such as at least MUC5B and MUC7, would also have to be controlled for completeness, which is another study limitation.”

We hope that this information will clarify the concern of the reviewer.

  1. Immunolocalization of MUC8 and TNFa:

One of the major drawbacks of this study is lack of control tissues to better understand the expression of MUC8 and TNFa in control male / female tissues. The authors will need to address this in the limitation of this study section.

We agree with the reviewer and have addressed this point in the limitation section as follows:

“Although we can verify the function of the antibodies used against MUC8 and TNFa with control tissues as stated, it is of course a major drawback of our study that there is no control tissue for the examination of the stones to directly compare the expression of the two proteins with male and female tissue.”

  1. Line 392: ‘4.1. Tissues, dacryoliths and saliva’

Is this a typing error? This is confusing, did the authors collected dacryoliths (naso-lacrymal stones) or sialoliths (salivary stones), did the authors performed expression studies using lachrymal stone or salivary stone?

  1. Though the sample size is very small, the authors observed higher MUC8 expression in females, this is a very interesting and novel observation. However, epidemiological evidence show either a higher prevalence in males or equal prevalence in male and females (https://www.nature.com/articles/sj.bdj.2014.1054 ). Do the authors observe more severity of the salivary stones in females? is higher MUC8 / TNFa expression related to the weight/size of the stone? It will be informative for the reader to include the stone weight/size in table 1. What is the status of salivary TNFa concentration in samples in table 2.

The questions asked by the expert are interesting, but unfortunately, they cannot be directly addressed in practice yet. Patients with salivary stones almost always consult a physician only when the suffering pressure of the disease is high. In Germany, moreover, the initial assessment usually takes place via a general practitioner in private practice, who then recommends that patients suck drops and often change their dietary habits. This can mean that it takes longer for patients to end up with an ENT specialist. However, the level of suffering can also vary extraordinarily between the sexes on an individual basis. For example, sometimes you have patients with very small stones and a high level of suffering and, on the other hand, patients with very large stones who actually only come because they feel and palpate the stone and it worries them, without them having an enormous level of suffering. In summary, we can state that we cannot say that we have observed a higher severity of salivary stones in females. Also, we can clearly say that the level of MUC8/TNFa expression does not correlate with the weight/size of the stone. We always measured the same amount of stone tissue in the ELISA and correlated it with each other in the analysis. There are clear subindividual variations here. It was only obvious that women on average have higher concentrations than men. Because of the above, we did not measure or weigh the stones and therefore cannot integrate the data. Unfortunately, we did not take this into account in the planning at the time. We also did not perform ELISA studies for TNFa, so we cannot add these data either.

"No statement can be made that women have more severity of the salivary stones. The level of suffering can vary extraordinarily between the sexes on an individual basis. For example, sometimes you have patients with very small stones and a high level of suffering and, on the other hand, patients with very large stones who actually only come because they feel and palpate the stone and it worries them, without them having an enormous level of suffering. Also, we can clearly say that the level of MUC8/TNFa expression does not correlate with the weight/size of the stone. We always measured the same amount of stone tissue in the ELISA and correlated it with each other in the analysis. There were clear subindividual variations here. It was only obvious that women on average have higher concentrations than men."

"Another study limitation is that we neither measured nor weighed the stones during surgical removal. For this reason, no information can be given on this. We also did not perform ELISA studies for TNFa, so we cannot add these ELISA data either."

  1. Figure 3 describes the dot blot of MUC8 expression in 24 salivary sones. I did not find any information for these 24 samples in the manuscript. The authors need to mention the sample details as a table. In addition, a densitometric analysis with respect to a control gene is required to achieve a statistical conclusion.

We apologize that this information was not clearly shown. As of note, we had already  provided the following information on the 24 patients suffering from stone disease  in the first version of the manuscript. We would like to guide the reviewer to the appropriate section "Sialoliths" in Materials and Methods. There, the 24 patients are mentioned. We very much hope that this clarifies the comment. In order of having additional remarks, we will be glad to state a response. Thank you.

Moreover, we would like to tell the reviewer, that an additional table would not add significant value to the manuscript, which can be also very confusing. We have performed densiometry, which we add here in the point-by-point response as a figure for the reviewer, but we feel that this information will not bring an additional benefit to our results.  Furthermore, the dot blot analysis is a very crude and easy method for protein detection. The more intense each dot is, the more MUC8 is occurred in the appropriate sample. However, in order to be able to make a more precise statement about the protein content, we specifically performed the MUC8 via the ELISA method so that the protein concentration could be determined quantitatively. The dot blot served us as a very simple detection method only for a pilot orientation to perform more precise experiments, e.g., with ELISA.

We added this in Materials and Method as follows:

"The dot blot served us as a very simple detection method to guide whether MUC8 was detectable in all stones."

Densiometric evaluation of dot blots.

We are aware that this is not 100% compliance with the request of the reviewer, but we believe that protein detection by ELISA is the clearly superior method. We hope that this answer has addressed the concern of the reviewer.

This manuscript is a resubmission of an earlier submission. The following is a list of the peer review reports and author responses from that submission.